



# Quantifying the effects of urban green space on water partitioning and ages using an isotope-based ecohydrological model

Mikael Gillefalk[1,2], Dörthe Tetzlaff[2,3], Reinhard Hinkelmann[1], Lena-Marie Kuhlemann[2,3], Aaron Smith[2], Fred Meier[4], Marco P. Maneta[5], Chris Soulsby[1,2,6]

[1] Chair of Water Resources Management and Modeling of Hydrosystems, Technische Universität Berlin, Gustav-Meyer-Allee 25, 13355 Berlin, Germany
[2] Department of Ecohydrology, Leibniz Institute of Freshwater Ecology and Inland Fisheries, Müggelseedamm 310, 12587 Berlin, Germany
[3] Department of Geography, Humboldt University of Berlin, Rudower Chaussee 16, 12489 Berlin, Germany
[4] Chair of Climatology, Technische Universität Berlin, Rothenburgstraße 12, 12165 Berlin Germany
[5] Regional Hydrology Lab, University of Montana, 32 Campus Dr., MT 59812, Missoula, USA
[6] Northern Rivers Institute, University of Aberdeen, St. Mary's Building, Kings College, Old Aberdeen, AB24 3UE, Scotland

*Correspondence to*: Mikael Gillefalk (mikael.gillefalk@tu-berlin.de)

**Abstract.** The acceleration of urbanisation requires sustainable, adaptive management strategies for land and water use in cities. Although the effects of buildings and sealed surfaces on urban runoff generation and local climate are well known, much less is known about the role of water partitioning in urban green spaces. In particular, little is quantitatively known about how different vegetation types of urban green spaces (lawns, parks, woodland etc.) regulate partitioning of precipitation into evaporation, transpiration and groundwater recharge; and how this partitioning is affected by sealed surfaces. Here, we integrated field observations with advanced, isotope-based ecohydrological modelling at a plot scale site in Berlin, Germany. Soil moisture, sap flow, together with stable isotopes in precipitation, soil water and groundwater recharge, were measured over the course of one growing season under three generic types of urban green space: trees, shrub and grass. Additionally, an eddy flux tower at the site continuously collected hydroclimate data. These data have been used as input and for calibration of the process-based ecohydrological model EcH$_2$O-iso. The model tracks stable isotope ratios and water ages in various stores (e.g. soils and groundwater) and fluxes (evaporation, transpiration and recharge). Green water fluxes in evapotranspiration increased in the order shrub (381 ± 1 mm) < grass (434 ± 21 mm) < trees (489 ± 30 mm), mainly driven by higher interception and transpiration. Similarly, ages of stored water and fluxes were generally older under trees than shrub or grass. The model also showed how the interface between sealed surfaces and green space creates edge effects in form of "infiltration hot spots". These can both enhance evapotranspiration and increase groundwater recharge. For example, in our model, transpiration from trees increased by ~50 % when run-on from an adjacent sealed surface was present and led to groundwater recharge even during the growing season, which was not the case under trees without run-on. The results form an important basis for future upscaling studies by showing that vegetation management needs to be



considered within a sustainable water and land use planning in urban areas to build resilience in cities to climatic and other environmental change.

**List of abbreviations**

| | |
|---|---|
| DWD | German weather service |
| $E_I$ | Interception evaporation |
| $E_S$ | Soil evaporation |
| ET | Evapotranspiration |
| KGE | Kling-Gupta efficiency |
| LAI | Leaf area index |
| LHS | Latin hypercube sampling |
| LID | Low impact development |
| NSE | Nash-Sutcliffe efficiency |
| P | Precipitation |
| RMSE | Root mean square error |
| SUEO | Steglitz urban ecohydrology observatory |
| SWC | Soil water content |
| T | Transpiration |
| TDR | Time domain reflectometry |
| TU | Institute of technology |
| UCO | Urban climate observatory |
| UHI | Urban heat island |

## 1 Introduction

Global urbanization is dramatic; in 2018, 55 % of the world's population was living in cities and by 2050 this is predicted to be 68 % (United Nations, 2019). This intensified urbanization presents many challenges for both urban-dwellers and policy makers. Cities have very different effects on hydrological partitioning compared to rural areas, with adverse effects on the water balance and water cycling. In rural areas, precipitation infiltrates into the ground and is either lost as evapotranspiration (green water) fluxes from vegetated areas, or groundwater recharge and surface runoff (blue water

fluxes). In cities, a larger portion of rainfall runs off sealed surfaces, creating flash floods, reducing groundwater recharge as well as degrading stream and river ecology (Paul and Meyer, 2001; Walsh et al., 2005). Consequently, green spaces and low impact developments (LIDs, Golden and Hoghooghi, 2017; Lim and Welty, 2018) are being increasingly used in urban landscape planning, aiming to retain water longer in built-up areas by reducing stormwater runoff, and increasing infiltration and groundwater recharge. This also contributes to mitigating the urban heat island effect (UHI, e.g. Peng et al., 2012) by

increasing evaporative cooling through latent heat fluxes (Bowler et al., 2010; Konarska et al., 2016). Such roles of urban greenspace are getting more attention from policy makers and the general population in terms of building resilience to climate change, which is expected to further amplify the severity of UHI effects in future. This can be seen as part of a



broader paradigm shift from viewing urban rainfall as to be something to remove as quickly as possible, to becoming a resource that needs to be stored and managed for multiple benefits (Gessner et al., 2014). These include environmental as

well as cultural and aesthetical benefits. Urban vegetation also helps removing pollutants in runoff, e.g. through the use of bioswales or rain gardens (Jamali et al., 2020), reduces noise, reduces air pollution by trapping pollutants (Nowak et al., 2006), increases biodiversity and gives recreational benefits. Green spaces are therefore crucial to help to actually expand urban areas, enabling more people to congregate and live in cities, while mitigating the effects of urbanization itself (Golden and Hoghooghi, 2017).

While the need for urban green spaces has been acknowledged for some time, little is known about the quantitative effects of such measures on water partitioning (Bonneau et al., 2017; Bonneau et al., 2018). In a broader sense, the effects are assumed to correspond to the goals of green spaces, namely enhancing infiltration, groundwater recharge and transpiration. But except for infiltration, quantitative knowledge on the quantities of water that is partitioned to evapotranspiration and groundwater recharge is still lacking (Kuhlemann et al., 2020a). Moreover, differences in vegetation types, and even different species, are

expected to lead to differences in water partitioning (e.g. Konarska et al., 2015; Muñoz-Villers et al., 2019). For example, trees have a higher interception capacity than grass; they also have a higher water demand and the capability to draw water from deeper soil layers, potentially increasing transpiration. However, in an urban setting, this might be different compared to a rural setting, due to, for example, shading by tall buildings, manmade subsurface structures (Bonneau et al., 2017) and the oasis or clothesline effect because of surface heterogeneity which affects evapotranspiration by advection (Oke, 1979;

Hagishima et al., 2007). Also, the UHI effect, increasing soil evaporation, could be expected to affect grass more than trees, as trees limit soil evaporation (Ellison et al., 2017).

Over the past decade, physically-based ecohydrological models have become increasingly used to explicitly conceptualise water partitioning by different vegetation types and quantitatively disaggregate green and blue water fluxes (Brewer et al., 2018, Fatichi et al., 2016). More recently, such models have been adapted to incorporate the specific effects of urban areas

(e.g. shading by tall buildings, heat storage and release by urban infrastructure, modified wind field in urban canopies etc.) and the use of ecohydrological models in urban setting is beginning to increase (Meili et al., 2020; Revelli and Porporato, 2018; Shields and Tague, 2015). EcH$_2$O is a process-based, distributed ecohydrological model developed by Maneta and Silverman (2013). EcH$_2$O couples energy balance and water balance modules, and integrates them with a biomass module that explicitly conceptualises the soil-plant-atmosphere-continuum. A recent development of EcH$_2$O has led to the EcH$_2$O-

iso model, which includes an isotope tracking routine. This allows the transformation of the stable isotopes of water in precipitation to be simulated as it is routed through ecohydrological systems and is affected by evaporative fractionation and mixing processes (Kuppel et al., 2018b). Stable isotopes have been widely used as tracers in catchment hydrology at multiple scales to better understand water flux-storage interactions and estimate water ages (Kendall and McDonnell, 1998), including incorporation in models (Birkel and Soulsby, 2015), and have recently been used to aid understanding of plant

water sources (e.g. Muñoz-Villers et al., 2019; Oerter et al., 2019; Geris et al., 2015). Despite some early studies (Harris et al., 1999, Wilcox et al., 2004), the application of stable isotopes in urban hydrology has been relatively limited, and their use



has been identified as a major research opportunity for solving applied urban water problems (Ehleringer et al., 2017). Here, we seek to follow recent successful applications of the EcH$_2$O-iso model to quantify ecohydrological fluxes and water ages in various settings (e.g. Smith et al., 2019; Smith et al., 2020a; Knighton et al., 2020) through an application to urban green spaces in Berlin, the capital and largest city in Germany.

Berlin is already one of Europe's largest cities, and like many others, it continues to rapidly grow. Between 2014 and 2019, Berlin's population grew by 40,000 people per year and reached 3,670,000 inhabitants (Amt für Statistik, 2020a). Moreover, climate change is affecting Berlin by changing precipitation patterns and near-surface atmospheric moisture (Langendijk et al., 2019; Kuhlemann et al., 2020a), increasing air temperatures and significantly increasing heat wave occurrence and duration (Fenner et al., 2019a), thereby exacerbating the UHI effect under hot weather episodes (Fenner et al., 2019b). The policy makers of Berlin are aware of these issues and are implementing mitigation measures. Berlin has a high coverage of green and blue space (30 %, Amt für Statistik, 2020b) and actors developing new areas are required to integrate some green space, not the least in order to ensure local infiltration of storm water.

Our overall aim is to model how different vegetation types comprising urban green spaces affect ecohydrological water partitioning. As a step towards providing an evidence base to achieve this, we focus on the Urban Ecohydrological Observatory at Steglitz (SUEO), in the south of Berlin, Germany. At the site, we identified three different vegetation types: trees (larger trees, on average 20 m high), shrub (bushes and smaller trees, max. height around 5 m) and grassland. The emphasis of research so far at the site has been on data collection conducted by Kuhlemann et al. (2020b), providing crucial input, calibration and validation data sets to test the EcH$_2$O-iso model in an urban setting for the first time. Using the model, we sought to increase the process-based understanding of urban green spaces; specifically, we want to answer the following questions:

**Research questions:**

1. What are the effects of different vegetation types on the ecohydrological partitioning of urban water (interception, infiltration/recharge, soil evaporation, transpiration)?

2. Can we use information on stable water isotopes to constrain soil water ages and resulting partition fluxes?

3. How do sealed surfaces affect infiltration and water partitioning in adjacent green space? Are there any edge effects?



## 2 Methods and material

### 2.1 Site description

The Steglitz Urban Ecohydrological Observatory (SUEO, Latitude: 52.457232, Longitude: 13.315827, at 48 m.a.s.l) in Berlin-Steglitz is located in the grounds of TU Berlin's Institute of Ecology and coincides with the Rothenburgstrasse site of the Urban Climate Observatory (UCO) operated by the Chair of Climatology at TU Berlin (Fig. 1). The UCO Berlin provides long-term data of atmospheric observations and is an open infrastructure for integrative research on urban weather, climate, and air quality (Scherer et al., 2019). There is an eddy flux tower on site (height 40 m), which has been operating since June 2018.

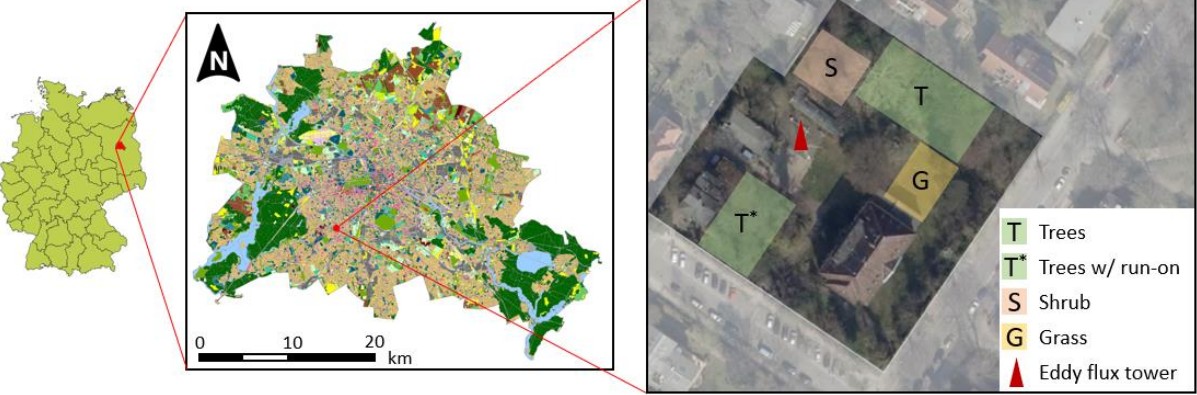

**Figure 1: From left: Germany, Berlin and the SUEO with vegetation plots and eddy flux tower. Data source basemaps: Senstadt 2015, Senstadt 2018.**

The size of the study site is 7800 m². On the premises, there is extensive green space, but there are also three buildings and one greenhouse. Parts of the study site are sealed (asphalt) or partially sealed (packed gravel). The vegetation is highly heterogeneous but representative for Berlin's (and other temperate-zone cities) urban vegetation and can be divided into three generic land cover types: trees, shrub and grass. The trees represent areas with larger trees around 20 metres high (example species *Fagus sylvatica*, *Platanus x hybrida*, *Frondibus ulmi*, *Quercus* and *Acer platanoides*), while the shrub areas contain younger trees, max height around 5 m (example species*, Acer platanoides, Acer pseudoplatanus*) and shrubs (*Clematis, Hedera helix, Rubus armeniacus,* for more details see Kuhlemann et al., 2020b). Across the site, the built area comprises 17 % buildings and 16 % sealed or semi-permeable parking areas or path. The remainder is urban green space with 39 % trees, 16 % is grassland and 7 % shrub.  The soil of the site is characterised by freely draining sands with an upper organic horizon from 0–20 cm. The subsurface is heavily impacted by human activities, and in places has an added layer of up to 50–180 cm of debris (e.g. from fill or demolished structures), sandy materials and a humus layer on top of the naturally occurring subglacial till (Bornkamm and Köhler, 1987). Groundwater levels are at ~10–15 m below surface (Senstadt, 2010). Berlin's climate is between a warm-summer humid continental climate and a temperate oceanic climate (Köppen classification: Cfb and Dfb, Beck et al., 2018). The long-term (1981–2010) mean annual precipitation is 591 mm and



temperature 9.7 °C (DWD 2020). The precipitation is almost equally divided between winter and summer, the former
characterised by more frequent lower intensity longer frontal, the latter dominated by sporadic, more intensive, convective
rainfall.

This study focuses on the period from 27 March to 30 November 2019 (approximating the growing season in Berlin).
Rainfall for the 12 months up until the end of the study (December 2018–November 2019) was 528 mm; ∼10% lower than
the long-term average. However, the preceding 12 months included the 2018 European drought (December 2017–November
2018), during which there was only 348mm of precipitation, 40 % lower than the long-term mean.

## 2.2 Data availability

Inputs needed to drive the ecohydrological modelling are hydroclimate data, including precipitation, short-wave radiation,
long-wave radiation, wind speed and humidity. The hydroclimate data were measured by the on-site eddy flux tower for the
period 1 June 2018 until 30 November 2019 and by another urban eddy flux tower (10 m above roof) at the UCO site TU
Berlin Campus Charlottenburg located 6 km north of the site for the period 1 March 2017 until 31 May 2018. Turbulent
sensible and latent heat fluxes were derived from an eddy-covariance system installed at top of flux towers, which combines
an open-path gas analyser and a three-dimensional sonic anemometer-thermometer. The software EddyPro (Version 6.2.1)
was used to quality control the raw data and to calculate turbulent fluxes from 20-Hz time series over 30-min intervals, and
then further averaged to hourly mean values. The data from the eddy flux tower had very few (<1 %) missing values during
the calibration period. Precipitation was measured at the German Weather Service (DWD) station located approximately 1
km south-west of the site which records essentially the same rainfall. This was chosen to get continuous precipitation data
series beginning in 2017 until the end of 2019, as the tower on site was installed in June 2018.

Measurement of soil water content (using Time Domain Reflectometry; TDR) is ongoing below trees, shrub and grass at
three depths (two sensors at each depth: 10–15 cm, 40–50 cm and 90–100 cm) since 24 March 2019. Measurement of sap
flow by sap velocity sensors measuring temperature differences between heated sensors is ongoing on six representative
urban trees since end of March 2019. Soil sampling and consequent stable water isotope analysis of the bulk soil water was
performed monthly from April to September and in November of 2019. Data on stable isotopes in precipitation were
measured at the IGB location in Berlin-Friedrichshagen (daily from August 2018 until March 2019) and from the SUEO in
Berlin-Steglitz (daily from March 2019 onwards). To hindcast an extended timeseries of stable isotopes in precipitation for
the model spin up, data from Zittau, 200 km southeast of Berlin (monthly from March 2017 until August 2018) were used
(IAEA/WMO, 2020). For a detailed description of measurements at the SUEO, see Kuhlemann et al., 2020b.

The modelling period followed on from the 2018 drought which continued through the winter of 2018/19. Despite the low
precipitation input, soil moisture levels at Steglitz were moderately high (∼15–25 %) at 10–15 and 40–50 cm at the
beginning of the measurement period in March 2019, with the sub-soil being wetter under grass (∼30 %) and shrub (∼25 %),
and lower under trees (∼10 %, Fig. 2). The early part of spring 2019 was relatively dry, with gradually increasing
temperatures and a gradual drying of the soils. The heavy convectional rain occurring at the beginning of June re-wetted the

upper soil at all plots. Drying again characterised a period with little rain through the rest of June and most of July, before small events in late July resulted in some partial re-wetting in the upper soils. However, it was not until a rainy period in late September/early October that upper soil moisture levels returned to those observed in spring. However, it was notable that at

all three plots, soil moisture level at 90–100 cm declined throughout, indicating no percolation to depth.



**Figure 2: Precipitation, $\delta^2H$ isotope in precipitation, air temperature at 2 m and soil water content (SWC) under the three vegetation types at the SUEO from March–November 2019. All hourly values except for $\delta^2H$ isotopes where samples were**
**collected daily. The SWC values are expressed as mean (line) and range (shaded area) of the two sensors at each depth.**

**2.3 Model description and set-up**

EcH$_2$O-iso is a process-based, tracer-aided ecohydrological model integrating energy balance, water balance and vegetation dynamics in the form of an explicit representation of associated biomass production (Fig S1). The basic structure and parameterisation of the EcH$_2$O model is described in detail by Maneta and Silvermann (2013), whilst the isotope and water
age module (EcH$_2$O-iso) is fully explained by Kuppel et al. (2018a). The energy balance component is based on a flux-



gradient similarity approach. It uses solar and longwave radiation reaching the canopy layer to simulate latent heat, sensible heat and net radiation, while latent heat, sensible heat, net radiation, ground heat flux and latent heat of snow melt are simulated at the surface level. While being driven by solar and longwave radiation, the energy balance partitioning is controlled by air temperature, relative humidity and wind speed. The water balance component conceptualises interception in the vegetation canopy through leaf area index (LAI) and a parameter for storage capacity on the leaf (m of storage per unit LAI). Infiltration is estimated using the Green-Ampt model. The subsurface is conceptualised as three layers, with water redistributed following infiltration using a gravitational drainage model, driven by the exceedance of field capacity and the vertical effective hydraulic conductivity. Vertical redistribution upwards occurs when deeper soils are saturated. Lateral soil water movement occurs in layer 3 when saturation is above field capacity. Water movement is based on a computationally efficient kinematic wave approximation, driven by the steepest-slope approach. Infiltration excess and exfiltrated water (fully saturated soils) are routed to the next downstream cell as surface runoff. Evaporation is drawn from the upper soil layer, while transpiration is drawn from soil layers as a function of the vegetation rooting parameter. The soil cover can be partially or fully sealed and does then allow less or no infiltration or soil evaporation. The isotope module of EcH$_2$O-iso tracks the composition of stable isotope ratios ($\delta^2$H and $\delta^{18}$O) and water ages of precipitation through the model domain. This incorporates evaporative fractionation of soil water in the upper layer. Mixing between soil layers was estimated using the completely mixed storage assumption (Kuppel et al., 2018b).

The model domain for the SUEO was divided into 10×10 m grid cells (Fig 3a) and the model was run with an hourly time step. The entire modelling period was set from 1 March 2017 to 30 November 2019, allowing a 2-year spin-up period from 1 March 2017 to 27 March 2019. The calibration period was 28 March 2019 to 30 November 2019.

The vegetation was divided into three categories: trees, shrub and grass. By virtue of measurements under different vegetation types, we assumed different soil characteristics following the vegetation division. The soil division was also motivated by the highly heterogeneous character of urban soils (Mao et al., 2014).

As biomass production was not our focus, we turned off vegetation dynamics to reduce the number of parameters. As such, the LAI was constant over the year for each vegetation type, calibrated using literature values of growing season leaf area index.





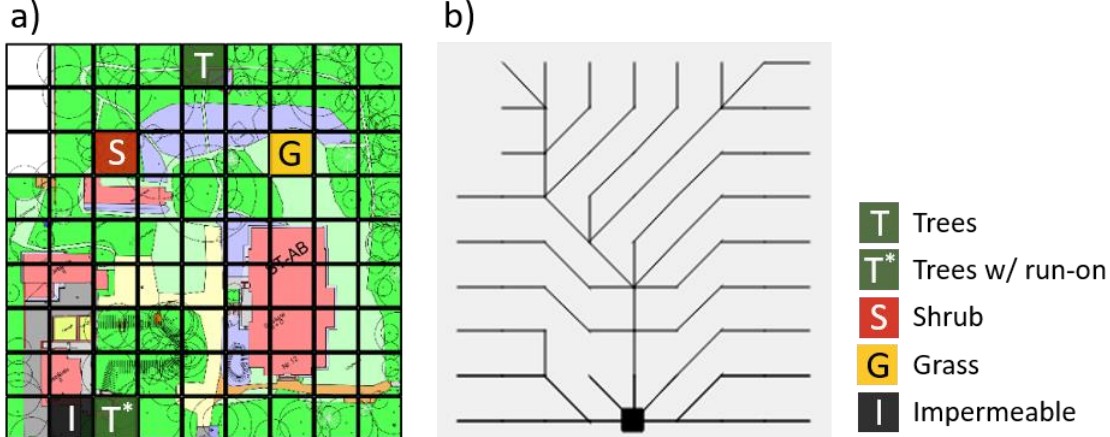

Figure 3: Model domain for SUEO (a) Model grid with impermeable (black), trees (with and without run-on, green), shrub (red) and grassland (yellow) cells marked, (b) Local drainage network with the outlet in the lower middle cell. Data source basemap (a): TU Berlin, 2018.

## 2.4 Calibration and validation

To calibrate the model, we used the Nash-Sutcliffe efficiency (NSE, Nash and Sutcliffe, 1970) and the Kling-Gupta Efficiency (KGE, Gupta et al., 2009) as objective functions. The Kling-Gupta efficiency takes correlation, variability bias and mean bias into account (Eq. 1).

$$KGE = 1 - \sqrt{(r-1)^2 + \left(\frac{\sigma_{sim}}{\sigma_{obs}} - 1\right)^2 + \left(\frac{\mu_{sim}}{\mu_{obs}} - 1\right)^2} \qquad (1)$$

where r is the linear correlation between the observations and the simulations, $\sigma_{obs}$ / $\sigma_{sim}$ the standard deviation in observations/simulations and $\mu_{obs}$ / $\mu_{sim}$ is the observation/simulation mean.

A sensitivity analysis using the Morris method (Morris, 1991; Sohier et al., 2014) was performed to establish which parameters are most relevant for soil water content. The Morris method uses a step-wise process, where parameters are varied independently by 50% of the parameter range and the degree of change in model output due to parameter variations is evaluated relative to the initial parameterisation. The initial parameters were randomized using latin hypercube sampling (LHS, McKay et al., 1979). The sensitivity was assessed using 80 trajectories and was evaluated using RMSE.

Multi-criteria calibration was conducted using soil water content in the upper two soil layer under trees, shrub and grass. We did not use data from the third layer as the deeper soil showed no measured percolation of water during the calibration period (Fig. 2). To constrain the parameter ranges and successfully calibrate the model we used an iterative process. Using subsets of the parameter ranges shown in Table 1, sets of 30,000–50,000 Monte Carlo runs were conducted to narrow the parameter ranges. After narrowing the ranges for all vegetation and soil types, the depths and soil layer thicknesses were calibrated for themselves. The single best values for depth and soil layer thicknesses were chosen and, as a final step, a set of 120,000 runs using the achieved narrower ranges was performed.





The best 20 runs based on the multicriteria calibration using KGE and NSE for soil water content in the upper two soil layers
were used for visualisation and calculation of essential output parameters. Tables S1 and S2 show the final calibration ranges
and the parameter values not part of calibration. To validate the model qualitatively, we compared measured and simulated
stable water isotopes ($\delta^2$H and $\delta^{18}$O) in the soil layers, radiative surface temperature with simulated soil skin temperature and
measurements of sap flow and simulated transpiration. The radiative surface temperature was calculated from measured
upwelling longwave radiation according to Stephan-Boltzmann law (Eq. 2):

$$T_{RS} = \left(\frac{R^{\uparrow}_{LW}}{\sigma}\right)^{\frac{1}{4}} - 273.15 \qquad\qquad (2)$$

where $T_{RS}$ is the radiative surface temperature (°C), $R^{\uparrow}_{LW}$ is the upwelling longwave radiation (W m$^{-2}$) and $\sigma$ is the Stephan-
Boltzmann constant (W m$^{-2}$ K$^{-4}$).

**Table 1: Initial parameter ranges of the soil and vegetation parameters used for trees, grass and shrub.**

| Parameter | Abbreviation | Calibration range | | |
|---|---|---|---|---|
| **Vegetation parameters** | | **Trees** | **Grass** | **Shrub** |
| Vegetation albedo [-] | $\alpha_{veg}$ | 0.1–0.2 | 0.1–0.2 | 0.1–0.2 |
| Leaf area index (LAI) [-] | LAI | 4.0–5.0 | 2.0–3.0 | 3.0–3.5 |
| Maximum stomatal conductance [ms$^{-1}$] | $gs_{max}$ | 0.0005–0.05 | 0.0005–0.05 | 0.0005–0.05 |
| Stomatal sensitivity to vapor pressure deficit | $gs_{vpd}$ | 1.0e-4–1e-3 | 1e-4–1e-3 | 1e-4–1e-3 |
| Stomatal sensitivity to light | $gs_{light}$ | 100–500 | 100–500 | 100–500 |
| Stomatal sensitivity to soil moisture content | $L_{WPC}$ | 1e-4–5e1 | 1e-4–5e1 | 1e-4–5e1 |
| Light extinction coefficient for the canopy [-] | $K_{beer}$ | 0.2–0.8 | 0.2–0.8 | 0.2–0.8 |
| Optimal growth temperature (° C) | $T_{opt}$ | 10–25 | 10–25 | 10–25 |
| **Soil parameters** | | **Trees** | **Grass** | **Shrub** |
| Total soil depth [m] | $D_{soil}$ | 1–7 | 1–7 | 1–7 |
| Thickness of 1st hydrological layer [m] | $D_{L1}$ | 0.15–0.35 | 0.15–0-3 | 0.15–0.35 |
| Thickness of 2nd hydrological layer [m] | $D_{L2}$ | 0.35–0.55 | 0.25–0.6 | 0.25–0.6 |
| Porosity [-] | H | 0.35–0.5 | 0.35–0.5 | 0.35–0.5 |
| Air-entry pressure head [m] | $\psi_{AE}$ | 0.15–0.55 | 0.15–0.55 | 0.15–0.55 |
| Saturated horizontal hydraulic conductivity [ms$^{-1}$] | $K_{EFF}$ | 1e-5–5e-2 | 1e-5–5e-2 | 1e-5–5e-2 |
| Exponential root profile [-] | $k_{root}$ | 0.1–10 | 5–20 | 0.5–10 |
| Brooks-Corey exponent [-] | $\lambda_{BC}$ | 2–5 | 2–5 | 2–5 |

## 2.5 Exploring the effect of sealed surfaces

Following the multi-criteria calibration, the model was also used to explore the effects of sealed surfaces on ecohydrological
partitioning. In the context of the study site, this allows quantification of increased runoff from impermeable surfaces and
"infiltration hotspots" at sharp interfaces between impermeable and permeable surfaces to be identified and explored (Voter
and Loheide, 2018). Of particular importance is an assessment of the implications of enhanced evapotranspiration by urban
green spaces as a result of this water subsidy. This was done by choosing an area in the lower part of the model domain (Fig.
3a), where precipitation falling on a sealed surface was routed along the topographic gradient to an adjacent tree-covered
area (Fig. 3b). This particular tree-covered area (from now on called trees with run-on) would therefore receive twice the
water compared to the tree-covered area without run-on used in the calibration process.



## 3 Results

### 3.1 Energy balance

As there were no measurements of soil skin temperature at the site, instead the modelled soil skin temperature values were compared to radiative surface temperature calculated from the measured upwelling longwave radiation by the on-site eddy flux tower (Figs. 4 and S2). The modelled soil skin temperature based on the 20 best runs followed the general pattern of radiative surface temperature well.

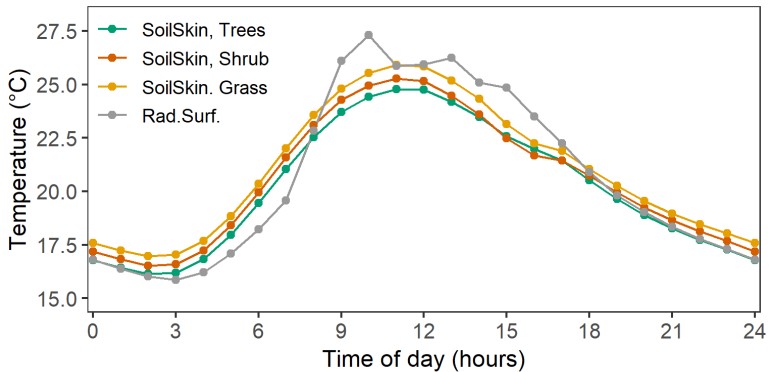

**Figure 4: Diurnal summer (June to August) temperature cycles. Modelled soil skin temperature (SoilSkin, average of 20 best runs) for all vegetation types (trees, shrub and grass) and calculated radiative temperature (Rad.Surf.) from upward longwave radiation.**

### 3.2 Soil water content

The model reproduced the soil water content in layers 1 and 2 reasonably well (Fig. 5). For trees and shrubs, the dynamics of layer 1 were generally captured. For the grassland, the model performed well for the April to early July period, but over-predicted re-wetting in late August and September. This might be due to changes in preferential flow, for example due to surface cracks, causing the sensors to miss the two rain events that the model simulated. The sensors below trees and shrub, however, did pick up the two events (Fig. 2). In layer 2, the model performed well throughout the period for shrub and grassland, while there was an under-prediction for trees. However, penetration of re-wetting fronts in early June and October was simulated (Fig. 5). This may in part reflect the calibrated depth of layer 2 being deeper than the soil moisture sensors. KGE values based on the single best multi-criteria run were for trees 0.67 and 0.37 for layers 1 and 2, respectively. For grass KGE values were 0.35 and 0.93 and for shrub 0.79 and 0.87, all for layers 1 and 2, respectively.

High heterogeneity in the soil moisture content was unsurprisingly apparent, and was particularly noticeable in the discrepancy between the two sensors in layer 3 under trees, but also under shrub (Fig. 2). This measured discrepancy is most likely due to differences in soil characteristics and this type of local heterogeneity was difficult to capture with the EcH$_2$O-iso model.



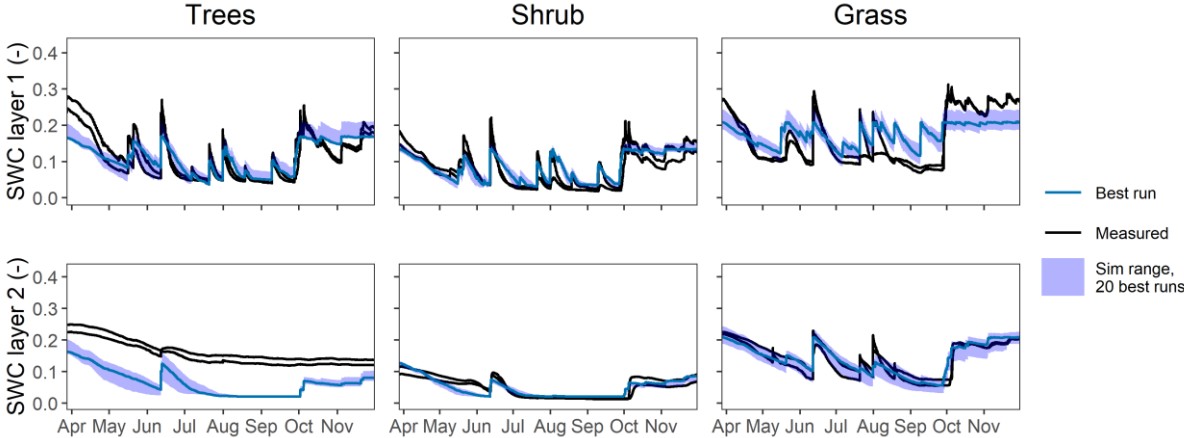

**Figure 5: Measured and simulated soil water content (SWC) in layers 1 and 2 under the trees, shrub and grass.**

### 3.3. Soil water isotopes

Although not calibrated on isotopes, the soil water isotope dynamics reproduced by EcH$_2$O-iso for layer 1 were captured very well with KGE values of 0.53, 0.73 and 0.68 for $\delta^2$H and 0.50, 0.68 and 0.61 for $\delta^{18}$O in trees, shrub and grass (Figs. 6 and S3). This indicates that the model provided a reasonable representation of the infiltration of effective precipitation and soil water mixing over the growing season. The effect of enriched precipitation in the summer, together with evaporative fractionation, raised both $\delta^2$H and $\delta^{18}$O ratios in all three profiles. These results add confidence that soil evaporation was being simulated quite well under each land cover. The influence of water from more depleted autumn rainfall was also evident in the more negative isotope ratios in October and November. In layer 2, dynamics were lessened in both measured and modelled values. The influence of the wetting front was also nicely captured, bringing more enriched water in spring and more depleted water in autumn (Fig. 6). However, modelled ratios were over-predicted in the summer for shrub and grassland. In layer 3, the modelled isotopic composition showed no change due to lack of penetration of wetting fronts over the modelling periods. This was consistent with the soil water content measurements (Fig. 2).





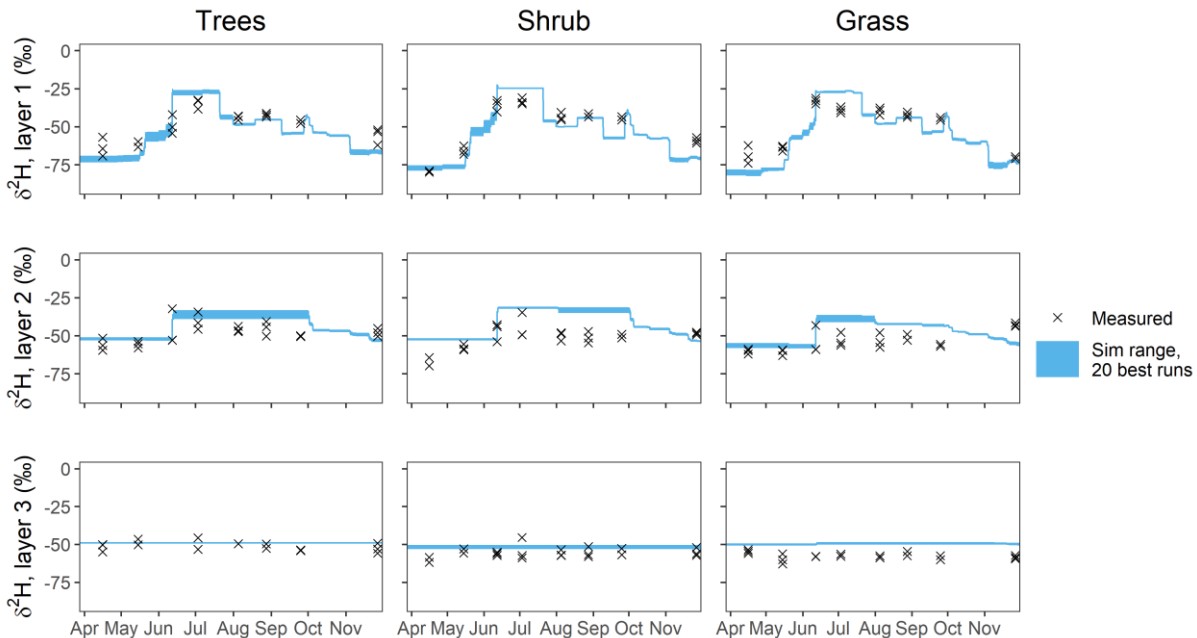

**Figure 6: Measured and simulated stable water isotope δ²H in bulk soil water. KGE values given for layer 1.**

### 3.4 Quantification of ecohydrological fluxes

The model allowed for the disaggregation of ecohydrological fluxes into interception, evaporation, soil evaporation and transpiration. We also conducted a "soft" validation comparing the modelled transpiration for the trees to measured sap flow (average of six trees) on site. Both sets of values were normalized and smoothed with a seven-day moving average to facilitate visual comparison (Fig. 7). In April, the transpiration was overestimated, probably as a result of the assumed constant growing season LAI. However, for the rest of the calibration period the fit was reasonable with slightly more 300 dynamic behaviour in the simulated values.

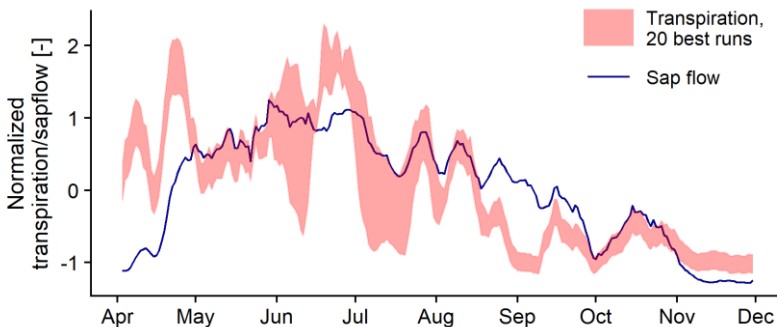

**Figure 7: Measured sap flux density (average of six trees, blue line) and simulated transpiration (red envelop). Both time series were normalized and smoothed with a seven-day moving average to facilitate visual comparison.**





For all three vegetation types, total evapotranspiration (ET) was highest between May and August, with a peak in June (not shown). Cumulative components of ET (Fig. 8 and Table 3) over the growing season showed that trees had the highest water loss at 489 ± 30 mm, followed by grassland (434 ± 21 mm) and shrub (381 ± 1 mm). This compares with the 352 mm of precipitation during the calibration period. Figure 8 shows the dominance of the transpiration fluxes for all vegetation types, though the uncertainty was high for trees in the late summer. Total transpiration decreased in the order trees > grass > shrub

with maximum transpiration rates of ~4 mm day$^{-1}$ for trees and ~3 mm day$^{-1}$ for grass and shrub. Evaporation of intercepted water was highest for trees, followed by shrub and grass. Soil evaporation was highest for grass, though it was low for all vegetation types in comparison to both interception evaporation and transpiration.

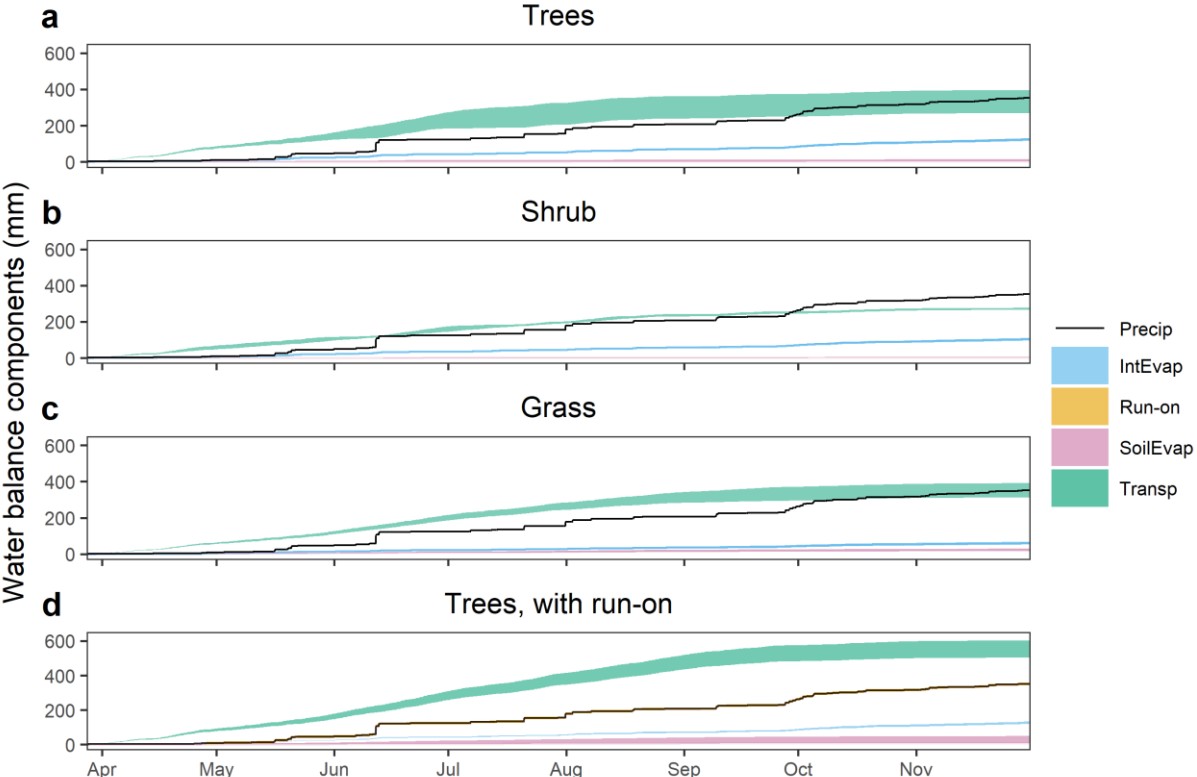

**Figure 8: Cumulative precipitation (Precip), evaporation from intercepted water (IntEvap), run-on, evaporation from soil water**
**(SoilEvap) and transpiration (Transp) for trees (a), shrub (b), grass (c) and trees with run-on from adjacent sealed surface (d).**

Scaling up to the whole area of the urban green space at the study site and taking the ratios of the three vegetation types (Sect. 2.1) into account but ignoring routing from sealed surfaces, total evapotranspiration was 287 ± 15 mm during the calibration period. The sum of evapotranspiration measured by the eddy flux tower on site was 328 mm. That the eddy flux tower measured a larger amount was expected as the model does not take into account evaporation from buildings or sealed

surfaces.





**Table 3: Modelled water balance components transpiration, interception evaporation, soil evaporation, total evapotranspiration (ET) and infiltration. Median and standard deviation of 20 best runs.**

| Veg. type | Transpiration [mm] | Intercept. evap. [mm] | Soil evap. [mm] | Total ET [mm] | Infiltration [mm] |
|---|---|---|---|---|---|
| Trees | 361 ± 31 | 124 ± 4 | 4 ± 3 | 489 ± 30 | 228 ± 4 |
| Shrub | 272 ± 3 | 107 ± 3 | 3 ± 1 | 381 ± 1 | 245 ± 3 |
| Grass | 353 ± 20 | 62 ± 5 | 22 ± 3 | 434 ± 21 | 290 ± 5 |
| Trees, with run-on | 543 ± 24 | 127 ± 4 | 13 ±12 | 686 ± 24 | 576 ± 4 |

### 3.5 Water ages

Tracking water ages within EcH$_2$O-iso showed that under all vegetation types soil water ages increased with depth (Fig. 9

and Table 4). Ages varied most dynamically in layer 1 – increasing to a few months over drier conditions and reducing to a few weeks when the system was wet after rainfall. In layer 2, the ages were older than six months when conditions were dry and younger than a few months after rainfall when the soils wetted up and wetting fronts penetrated.

Under grass, water penetration to layer 2 was most frequent and reduced water ages. In layer 3, no water percolation was observed or modelled during the study period, except for very small fluxes on three occasions in the simulations under grass.

As a result, ages in layer 3 were >2 years under shrub and grass and >3 years under trees.

Soil water ages under trees were older than under shrub and grass, and these differences increased with depth. This was expected from the lower effective rainfall under the tree canopy and the higher transpiration. Consequently, modelled ages in layer 1 were 2–4 weeks older under trees than under shrub and grass, 2–3 months older under layer 2 and 1–2 years older in layer 3. Differences between shrub and grass were within model uncertainty.

Water ages in transpiration increased in the order shrub (30 days ± 3) < grass (95 days ± 18) < trees (151 days ± 32) (Table 4). The differences were much smaller for soil evaporation, where the ages under shrub were 30 days ± 1, grass 29 days ± 2 and trees 43 days ± 2 (Table 4).




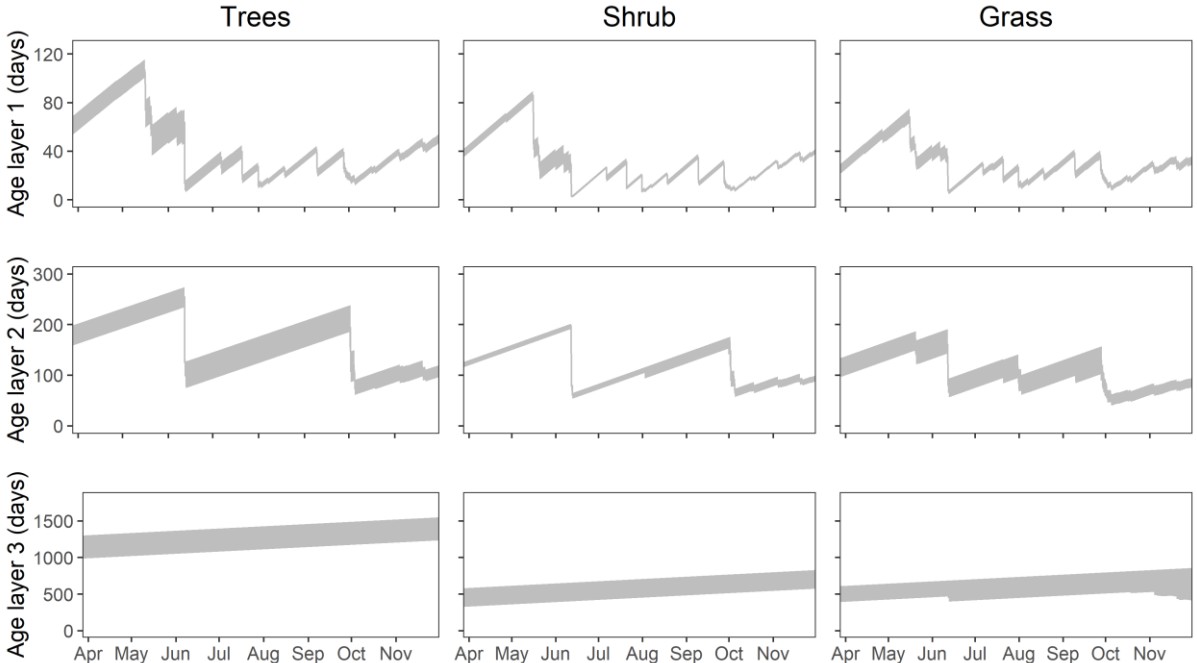

**Figure 9: Modelled water ages (days) in soil layers 1, 2 and 3 under trees, shrub and grass.**


**Table 4: Ages in soil water (three layers), transpiration, interception evaporation and soil evaporation (days) under trees, shrub and grass.**

| (days) | Trees | Shrub | Grass |
|---|---|---|---|
| Soil water, Layer 1 | 46 ± 26 | 30 ± 19 | 29 ± 14 |
| Soil water, Layer 2 | 171 ± 54 | 119 ± 38 | 103 ± 35 |
| Soil water, Layer 3 | 1258 ± 91 | 636 ± 128 | 607 ± 95 |
| Transpiration | 151 ± 32 | 30 ± 3 | 95 ± 18 |
| Soil evap. | 43 ± 2 | 30 ± 1 | 29 ± 2 |

### 3.6 Effects of sealed surfaces

The model also provided the opportunity to assess the effects of the addition of run-on from adjacent cells with sealed surfaces to the soil moisture regime and water partitioning (Fig. 3). An additional 352 mm of run-on subsidised soil moisture in the downslope cell. In the model, this increased transpiration by 50 %, soil evaporation by 225 % (albeit from an initially low volume) and total evapotranspiration by 40 % (Table 3 and Fig. 8a vs. 8d). Of course, this makes the extreme assumption that infiltration at the edge of the impermeable area is accessible over the grid square for evaporation and





transpiration. However, it likely indicates the upper limit of enhanced green water fluxes. Despite this, the enhanced runoff resulted in a recharge flux below the rooting zone of 20 mm. This contrasted with the behaviour of the three ecohydrological plots, where there was no recharge flux at all in the same period, as transpiration was higher than infiltration (Table 3).

## 4 Discussion

### 4.1. Tracer-aided modelling in urban settings

We successfully calibrated the ecohydrological, tracer-aided model EcH$_2$O-iso for three different generic types of urban green space: urban trees, shrub and grassland. We applied a multi-criteria calibration (Ala-Aho et al., 2017) to depth-dependent soil water content and qualitative validation using measurements of stable isotopes in bulk soil water. For the tree site, we could also check the robustness of the results and appropriate process representation using measured dynamics of sap flow as a proxy for transpiration and compare it to simulated transpiration. For all three vegetation types, we could
compare radiative surface temperature (calculated from measured upwelling longwave radiation) to simulated soil skin temperature.

A challenge for improving the fit of modelled soil water content was the fact that the measurements unavoidably were point measurements while the model output provides values integrated over depth-dependent layer averages. Also, there was inevitably considerable heterogeneity in soil characteristics, as expected when dealing with urban soils (Mao et al., 2014),
and as evidenced by the large discrepancy between the two sensors at the deepest layer under trees. Additionally, there were two rain events in August and September that were not picked up by the two shallowest sensors under grass (but by the sensors under trees and shrub), possibly because of change in preferential flow due to, for example, temporary clogging of pores or surface cracks in the top soil.

### 4.2 What are the effects of different vegetation types on the ecohydrological partitioning of urban water?

Results showed both expected and surprising differences in water partitioning during the study period. Water use in the urban green spaces increased in the order shrub (381 ± 1 mm) < grass (434 ± 21 mm) < trees (489 ± 30 mm). Whilst it is expected that the water use of trees would be greatest, it was surprising that modelled water use by shrub was lower than grass (Fig. 10). Disaggregation of water partitioning showed that whilst interception was highest in trees and shrub, trees and grass transpired around 30 % more than shrubs. This can be explained by a higher interception capacity of shrubs compared
to grass and by a deeper root depth (conceptualised through the K-root parameter) of trees compared to shrub to sustain transpiration (cf. Schenk and Jackson, 2002). It is likely that there is a slight overestimation of transpiration very early in the growing season for all vegetation types as a result of a fixed LAI, (cf. Fig. 7 for the trees) which is similar to the results of Douinot et al. (2019). Soil evaporation was highest under grass, due to the lower LAI and the deep shading under the trees and shrub canopies, but this remained only a small part of the total evapotranspiration losses (5 %).



A potential explanation for these somewhat unexpected results is that prevailing soil moisture conditions during the study period, especially in the upper soil layers, were very dry for prolonged periods. This may be a "memory effect" in the moisture regime of soil-vegetation systems from the severe drought of 2018 and low rainfall over the winter of 2018–19. Circumstantial evidence suggests that transpiration in trees and shrubs could have been suppressed during the study period. Kuhlemann et al. (2020b) showed that normalised transpiration for the tree plot at the SUEO did not always respond

proportionately to increases in atmospheric moisture demand (indexed through potential ET) implying the trees are conserving water and may be experiencing moisture stress at times (cf. Johnson et al., 2017). Additionally, the modelled transpiration response to the addition of run-on (Fig. 8) implies water limitation for trees inhibiting growth. Marchionni et al. (2019) showed how trees in Melbourne, Australia sustained transpiration using deeper groundwater which was around 3 and 4 m below the soil surface at their study site. At our site, however, this was an extremely unlikely strategy, as groundwater

depth is 10 to 15 below the surface (Senstadt, 2010). Pataki et al. (2011) reported maximum transpiration rates for irrigated street tress of ~2.5 mm day$^{-1}$, which is lower than our results for trees (~4 mm day$^{-1}$) but similar to our results for shrub (~3 mm day$^{-1}$). These results look reasonable taking into account that trees at the SUEO are larger, up to over 100 cm diameter at breast height, than those studied by Pataki et al. (2011), ~50 cm diameter at breast height.

**4.3 Can we use information on stable water isotopes to constrain soil water ages and resulting partition fluxes?**

Evaluating the model results using soil water isotopes proved to be insightful and increased the confidence that ECH$_2$O-iso was generally capturing processes adequately and producing "right results for the right reasons" (Kirchner, 2006). The generally good reproduction of the isotope dynamics in layer 1 supports this, as this simulation required that interception losses, soil evaporation and mixing processes in the most variable soil compartment and important ecohydrological interface were reasonably well-captured. The modelled response in layer 2 was somewhat less successful, though the dynamics of

isotopic change associated with penetrating wetting fronts were generally reproduced. This may reflect the full-mixing assumption and lack of differentiation between fast and slow water movement in each water layer in the version of EcH$_2$O-iso used (Kuppel et al., 2020). Also, the mismatch in scales between point measurements and a thicker, deeper modelled layer need to be considered in this context. Finally, the isotopes confirmed the lack of percolation of water to 1 m during the modelled period. Again, the similar measured isotope ratios to those modelled draining the profile as recharge in Layer 3

further adds confidence in the modelling results.

The isotope tracking within EcH$_2$O-iso also facilitates the estimation of water ages in the main storage components (i.e. soil layers) and fluxes. As such, this is insightful in terms of the time-variant changes in water ages in response to rainfall events. Under all vegetation types, soil water ages increased with depth, with more rapid turnover in layer 1, where ages were in the order of 4–6 weeks, depending on rainfall inputs. Ages increased to 4–6 months in layer 2 where percolation is limited, and

to 2–4 years in layer 3. Of course, the focus on the growing season likely resulted in a bias towards older ages than would be the case in the winter, where lower ET would cause greater percolation and recharge (Knighton et al., 2020).



**Figure 10: Conceptualisation of the simulation results: ecohydrological fluxes and ages over the growing season at the SUEO 2019. Transpiration (T, green), interception evaporation (E_I, light blue), soil evaporation (E_S, pink), precipitation (P) and surface runoff (dark blue).**

The older age of soil water under trees is consistent with the reduced turnover of water as a result of higher interception and transpiration withdrawals. These differences were most evident in layer 3 where differences in soil water ages between the grassland and shrub plots implied slightly younger water in the deeper soil layers for the latter (by 2–4 weeks), but these were still within model uncertainty (Table 4). The ages underline the importance of green water fluxes at the study site and



the sensitivity of recharge to vegetation cover (Douinot et al., 2019, Smith et al., 2020a). In layer 3 of the tree site, the water ages at the end of the simulation were in the order of ~2 years under grass and shrub and ~3 years under trees. In general, for Berlin, travel times through the unsaturated zone to groundwater are considered to be decadal (SenStadtWoh, 2019) which would be consistent with the low recharge and older modelled soil ages.

## 4.4 How do sealed surfaces affect infiltration and water partitioning in adjacent green space? Are there any edge
effects?

A striking result of the modelling application was the possibility to assess the potential significance of the permeability interface that occurs between sealed (i.e. impermeable) surfaces and urban green space in terms of enhancing infiltration and water subsidy to growing vegetation. The increase of transpiration by about 50 % on the model grid square with run-on from adjacent sealed surface compared to the tree-covered plot without run-on supports the point made above (Sect. 4.2) and
suggestion of Kuhlemann et al. (2020b) that the trees were water limited in the summer of 2019. Voter and Loheide (2018) showed that depending on the urban runoff design, an area with sealed surfaces can give rise to higher rates of deep drainage than without any sealed surfaces, partially explained by infiltration hotspots.

Of course, the results presented here need to be treated cautiously, the model assumes that all run-on is potentially available for transpiration in the receiving model 10x10 m grid square. In reality, the infiltration will be much more focused at the
permeability interface (Voter and Loheide, 2018). Nonetheless, it perhaps provides an assessment of the likely maximum effect, and it is likely that vegetation will increase rooting densities in areas where water and nutrients are available (Coutts et al., 1999). This finding has important implications for managing runoff from impermeable surfaces as part of LID, where one practise is to let water from for example a parking lot infiltrate in a ditch, with a little or a lot of vegetation (Roseen et al., 2006). The relative prioritisation of enhancing green or blue water fluxes, and the associated trade-offs in terms of
management objectives such as mitigating the UHI (Zölch et al., 2016) or enhancing groundwater recharge, could guide the design for functional green infrastructure (Berland et al., 2017).

## 4.5 Broader implications

The edge effects next to sealed surfaces showed that even if on average there is less infiltration in the urban landscape compared to a more natural setting, at certain locations there are likely to be artificially high rates of infiltration. This needs
to be considered when attempts are made to upscale models in urban areas, which typically is a significant challenge for urban hydrological modelling (Ichiban et al., 2018). To model larger areas at a fine scale might not be feasible but it would be important to identify where edge effects might occur and incorporate this, for example via parametrisation (Voter and Loheide, 2018) or by varying the mesh/raster size in those areas (Schubert et al., 2008). In a recent study, Smith et al. (2020b) showed how EcH$_2$O-iso in a rural catchment produced differences in ecohydrological process representation
depending on the chosen grid resolution, something that also needs to be tested in an urban setting.





When upscaling in space it may also be important to incorporate other components of the urban fabric, such as channels, cable and pipe trenches, leaky pipes etc (Bonneau et al., 2017). This type of infrastructure has been described as the "urban karst" (Bonneau et al., 2017) and might influence subsurface flows in a significant way. The "urban karst" is also relevant when planning to use urban green space for combating the urban heat island effect (Bowler et al., 2010). Infiltrating water

might be lost below the rooting zone and unavailable for trees to use for transpiration because of preferential flow paths created by for example cable and pipe trenches (Bonneau et al., 2017; Bonneau et al., 2018). Other factors also need to be taken into account when planning green space and LID measures. Increasing vegetated areas in a city creates or increases the need for irrigation and thereby amplifies the blue water footprint (Nouri et al., 2019; McCarthy and Pataki 2010). This is of course most relevant for cities with a dry and hot climate, but with ongoing climate change more cities will likely need to

irrigate to maintain their green infrastructure. However, Nouri et al. (2019) pointed out that a strategy to irrigate for optimal growth does not have to be essential. They instead suggested to maintain urban green with no (causing water stress) or very limited irrigation (e.g. when reaching a critical level of water stress). This also highlights the importance of choosing the right type of vegetation, for example choosing native species (Nouri et al., 2019) with a high rooting depth (Johnson et al., 2018), drought-resistant species (Muffler et al., 2020) or the right species for green walls (Prodanovic et al., 2019). Gómez-

Navarro et al. (2021) found that the right combination of trees and turf grass might be a good combination to optimize the mitigation of UHI while lowering the need for irrigation. A potential consequence of concentrated infiltration due to LID practices is that groundwater mounding can lead to the release of pollutants from the subsurface (Bonneau et al., 2018). All this highlights the need for extending the evidence base for rational decision making when it comes to improving water governance in urban areas (Kuller et al., 2017, Bach et al., 2020).

**5. Conclusion**

The ecohydrological model EcH$_2$O-iso successfully simulated dynamics in soil water content, soil water isotopes, energy balance and transpiration at the plot scale in an urban landscape in Berlin, Germany. The results showed that evapotranspiration fluxes increased from shrub < grass < trees and that soil water ages were higher under trees compared to shrub and grass, leading to lower and older groundwater recharge fluxes. We also could use the calibrated model to simulate

the effects of increased runoff from sealed surfaces unto a vegetation covered area where infiltration and transpiration was increased, indicating that trees at our study site were probably water limited during the summer of 2019. Our results show how important it is to choose the right type of vegetation in urban green spaces as there will often be a trade-off between increasing evapotranspiration in order to mitigate the urban heat island or increasing groundwater recharge and potentially baseflows. Depending on local needs, different types and density of vegetation should be chosen which will result in

different impacts. In order to further quantify the effects of different vegetation types on water partitioning in urban settings upscaling of the model in space, but also in time, will be needed in future. Future research also requires more



ecohydrological data on contrasting soil-vegetation systems in different urban settings; as we have shown, such data can be usefully enhanced by isotope-based methods.

## Code and data availability

The model code of EcH$_2$O-iso is available at https://bitbucket.org/sylka/ech2o_iso/src/master_2.0/. The data used are available from the author upon request.

## Author contributions

MG and CS were responsible for the formulating the overarching research goals and aims and prepared the original draft. FM curated the hydroclimate data. L-MK curated the measured data from the SUEO. AS led the work for implementing and
testing the model code. MG was responsible for performing the model runs, evaluation and visualization of the results. CS and RH acquired funding for the project and publication costs. CS and DT provided oversight for the research activity planning and execution. All authors contributed to discussion of results and the evolution of the written manuscript.

## Competing interests

The authors declare that they have no conflict of interest

**Acknowledgements**

The authors wish to thank the Einstein Foundation for financing the MOSAIC project, in which this study was performed. Contributions from CS were also funded by the Leverhulme Trust's ISOLAND project. The German Federal Ministry of Education and Research (BMBF) funded instrumentation of the Urban Climate Observatory (UCO) Berlin under grant 01LP1602 within the framework of Research for Sustainable Development (FONA; www.fona.de). MG is a collegiate of the
DFG Research Training Group Urban Water Interfaces (GRK 2032/2), which funded the publication. We thank Christian Marx for regular input during the modelling process, Lukas Kleine for help with figures and Ralf Duda for indispensable help with technical issues throughout the modelling process.

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
