# Peer review of "Quantifying the effects of urban green space on water partitioning and ages using an isotope-based ecohydrological model"

_Hydrology and Earth System Sciences, 2020_

## Author Comment (AC1)

**Comment on hess-2020-640**
**Anonymous Referee #1**

Referee comment with author replies on "Quantifying the effects of urban green space on water partitioning and ages using an isotope-based ecohydrological model" by Mikael Gillefalk et al., Hydrol. Earth Syst. Sci. Discuss., https://doi.org/10.5194/hess-2020-640-RC1, 2021

Author replies in red

**Overall comment**

This paper presents the results of an urban ecohydrological modelling exercise using the ECH2O-iso model. Growing season field measurements, including soil water content were used to calibrate the model. Qualitative validation was carried out by comparing measured and simulated water isotopes, surface temperature, and simulated transpiration to measured sap flow data. Overall, the paper was well organized and enjoyable to read. The clarity and simplicity of the figures was appreciated. I think there is room to improve the clarity of the methods with some additional details (see suggestions in specific comments below). As well, I recommend the authors try to make the model validation more quantitative (see suggestions in specific comments below). Lastly, I think the authors could place some additional focus on the urban design/management implications of the grass results, which suggest they are as important as trees for regulating green water fluxes.

We thank Reviewer 1 for her/his supportive comments and suggestions. In retrospect, we see that we could have been clearer in describing some of the methods used and gone further in validating the model. In addition, we agree that some of the management implications could have been better developed in the discussion. We are confident that we can address these issues on revision.

**Detailed comments**

48-9: Reverse order of this sentence, i.e., … by increasing infiltration and groundwater recharge, and thereby reducing stormwater runoff. OK

55: Replace 'removing' with 'to remove'. OK

57: Replace 'gives recreational benefits' with 'provides recreational opportunities'. OK

57-8: Remove '…to actually…' OK

63: Should be '… quantities of water that are partitioned…' OK

70-1: The point being made here is unclear. I recommend revising the wording for greater clarity. Will be revised

76: 'setting' should be plural. OK

81: Grammatical issue in this sentence – isotopes is plural, but sentence doesn't reflect that. Will be revised

96-7: Could the aggregated percentage of green and blue space be broken down for both? Will look in to this

Figure 1: Add a scale bar and north arrow to the zoomed in map of the observatory site. Will be added

144-5: Do the authors think that the drought conditions preceding the study could have impacted the results in any way? Yes, in fact we mention this in the discussion, lines 381-383: "This may be a "memory effect" in the moisture regime of soil-vegetation systems from the severe drought of 2018 and low rainfall over the winter of 2018–19. Circumstantial evidence suggests that transpiration in trees and shrubs could have been suppressed during the study period." We will emphasise this point further in the revision.

151: Missing words in this sentence - '… installed at the top of the flux towers…' Will be revised

150-7: What is the fetch (or footprint) of each tower? The tower on-site appr. 500-600 meters, the tower 6 km north of site appr. 700-800 m.
What proportion of the study period is each tower measuring convective fluxes that are representative of the observatory site? This will depend on wind direction of course. The tower on site should be measuring representative conditions at all times, regardless of wind direction. This therefore true for the calibration period, (as the tower on site went into operation the preceding summer).

Section 2.2.: Throughout this section it would be helpful to know the make/model of all the sensors that were used. Will be added.

161-4: How was soil sampled? How was water extracted? How was isotope analysis performed?
166: I recommend providing a brief overview of methods and then referencing Kuhlemann et al 2020b for more detail. We will add more details in the supplementary material to make our paper more standalone. But all details are available with Open Access in Kuhlemann et al. (2021), *HESS* (in the submitted manuscript this was referred as Kuhlemann et al. (2020b), since then the preprint has been accepted and published).

188-9: Is available moisture at the surface not used as well for partitioning available energy into convective fluxes? Yes, we will be clearer here.

198: How is the vegetation rooting parameter obtained? Is it based on field measurements of rooting depth? It is a calibration parameter, see Table 1, assuming an exponential root profile.

204: The calibration period is the period over which data were obtained from the site. Validation is qualitative. Is there 2020 data that can be used to validate? Thanks for this suggestion. Yes, there is SWC data but no soil isotope data. We have taken a look at this as per suggestion and results are the same as 2019, except for a poorer fit for Grass layer 1, where SWC generally is underestimated. We will include this validation evidence in the revision.

206: The soil division in the model is unclear to me. Were soil surveys used to characterize the spatial distribution of soil types? If not, was there any verification of the assumed soil characteristics? Many soil characteristics were used as calibration parameters and we made sure to stay within reasonable parameter ranges. But this was also based on previous soil surveys at the site. We will clarify this in the revision.

Figure 3b: Is the local drainage network storm sewers? Is there any channelized flow through the site?  The local drainage network is topographically defined and shows the direction that water would flow in the model in case of overland flow. In our study, the local drainage network is really only relevant for the exploratory study described in sections 2.5, 3.6 and 4.4. For the green spaces that are the focus of the study, there is no overland flow.

239: I think you could use 'ground-to-atmosphere' instead of 'upwelling'. We will clarify on revision.

247-9: How common are "infiltration hotspots" at sharp interfaces between impermeable and permeable surfaces? In my experience, these areas are usually quite compacted, thereby preventing infiltration. This is a good point and we will acknowledge this issue on the revision. However, in Berlin the sandy nature of the soil means that compacted areas are limited to areas with high footfall. That is not the case at the study sites, but we will acknowledge this limitation.

267-8: Were surface cracks visible to the field team? No, they weren't. Another interpretation, based on subsequent data collected at shallower depths, is that water did not reach down to the sensors at 10 cm, while in the model the top layer starts at the surface and therefore registers any infiltration into the soil. This will be added in the paragraph.

274-5: Are the authors referring to the measured or modelled SMC here? Layer 3 is not shown in Figure 2. It's not clear which data is being discussed. It is mentioned in the following sentence that we are referring to the measured SMC. We will rearrange the two sentences to make this clear.

299-300: I think the authors should try to provide some interpretation of the poor fit in late June, late Aug and early Sep. Thanks for this suggestion. Including the sap flow measurements was not meant to be a quantification of transpiration, what we wanted was a qualitative comparison of the variability. We have now also updated Figure 7 as per request of reviewer #2 (see below), adding an envelope showing the range of the measured sap flow for the individual monitored trees.

307-8: What do the authors mean by 'This compares with the 352 mm of precipitation during the calibration period.'? All the ET values are larger than P. Will be reformulated. "This can be compared to the 352 mm…"

318-20: Is there some way to estimate E from these surfaces to see if they account for the missing amount? Yes, with some assumptions about surface storage on buildings/sealed surfaces and evaporation rates, this would be possible but we would prefer to keep the focus of this paper on urban green space.

324-7: In layer 1, SWC is generally higher at the beginning and end of the study period with fluctuations (but lower baseline) in between. Why is the water age relatively high in April/May and Nov? Maybe the April soil water is old from the previous winter, but November receives a fair amount of precip. Some explanation of the distribution of ages at the beginning and end of the study period would be helpful. In fall/winter much less water leaves the compartments, less transpiration, less evaporation, and there is mixing. You do see that when there is rainfall, the ages go down, but not as extreme as in summer when old water continuously leaves the compartments. We will make this clearer in the revision.

362-5: Would the authors recommend more, denser SWC measurements in urban soils? That is one option if resources are available. Another idea would be to integrate the measurements over the soil profile and create virtual layers from that within the model domain.

391: Spelling error – trees. OK.

396: It's unclear what '… generally capturing processes adequately…' means. What were the criteria for evaluating this? Visual comparison.
On line 397, 'generally good reproduction' is used. Can the authors be more quantitative,

e.g., measured and modelled isotope values were within x-x % of one another? We have presented KGE values for layer 1, but did not for layers 2 and 3 as there are lower dynamics. In the revision we will look to use RMSE or ME as an alternative.
It's unclear what is meant by '… this simulation required…'. Will reformulate

399: Similar to the last comment, '… somewhat less successful…' seems too vague. Can the fit be assessed quantitatively? (As above)

399-400: I recommend being more explicit about the number of wetting fronts that the isotopic results reflected. We will clarify that in revision.

Figure 10: This is a nice conceptual figure. Thank you.

Section 4.4: It might be worth mentioning that sometimes preferential flow pathways can form along impermeable-permeable surface boundaries. This would move water away from the area and potentially make it unavailable for green or blue water fluxes. Something to explore in future work perhaps. [I see the authors make this point later on line 454-6… excellent].

Section 4.5: Could the authors comment further on the role of grass in promoting green water fluxes? Many sustainability-focused landscape designers seem to be moving away from grass (or lawns), but perhaps the findings of this study are an argument in their favour. We do mention the study by Gómez-Navarro et al. 2021, which stated that a combination of turfgrass and trees can be beneficial in combating UHI. But we can definitely elaborate further on this topic, with respect to, for example, differences in shading and irrigation needs between trees and grass.

**References**

Kuhlemann, L.-M., Tetzlaff, D., Smith, A., Kleinschmit, B., and Soulsby, C.: Using soil water isotopes to infer the influence of contrasting urban green space on ecohydrological partitioning, Hydrol. Earth Syst. Sci., 25, 927–943, https://doi.org/10.5194/hess-25-927-2021, 2021.

**Suggestion for new version of figure 7:**

---

## Author Response (AR1)

**Author response on hess-2020-640**

This document entails detailed responses on two referee comments as well as a summary of relevant changes made during revision of the manuscript, in that order.

**Comment on hess-2020-640**
**Anonymous Referee #1**

Referee comment with author replies (post-revision) on "Quantifying the effects of urban green space on water partitioning and ages using an isotope-based ecohydrological model" by Mikael Gillefalk et al., Hydrol. Earth Syst. Sci. Discuss., https://doi.org/10.5194/hess-2020-640-RC1, 2021

Author replies in red

**Overall comment**

This paper presents the results of an urban ecohydrological modelling exercise using the ECH20-iso model. Growing season field measurements, including soil water content were used to calibrate the model. Qualitative validation was carried out by comparing measured and simulated water isotopes, surface temperature, and simulated transpiration to measured sap flow data. Overall, the paper was well organized and enjoyable to read. The clarity and simplicity of the figures was appreciated. I think there is room to improve the clarity of the methods with some additional details (see suggestions in specific comments below). As well, I recommend the authors try to make the model validation more quantitative (see suggestions in specific comments below). Lastly, I think the authors could place some additional focus on the urban design/management implications of the grass results, which suggest they are as important as trees for regulating green water fluxes.

We thank Reviewer 1 for her/his supportive comments and suggestions. In retrospect, we see that we could have been clearer in describing some of the methods used and gone further in validating the model. In addition, we agree that some of the management implications could have been better developed in the discussion. We have addressed the reviewer's comments and revised the manuscript accordingly.

**Detailed comments**

48-9: Reverse order of this sentence, i.e., … by increasing infiltration and groundwater recharge, and thereby reducing stormwater runoff.
Changed

55: Replace 'removing' with 'to remove'.
Changed

57: Replace 'gives recreational benefits' with 'provides recreational opportunities'.
Changed

57-8: Remove '…to actually…'
Removed

63: Should be '… quantities of water that are partitioned…'
Changed

70-1: The point being made here is unclear. I recommend revising the wording for greater clarity.

Revised

76: 'setting' should be plural.
Changed

81: Grammatical issue in this sentence – isotopes is plural, but sentence doesn't reflect that.
Revised

96-7: Could the aggregated percentage of green and blue space be broken down for both?
We have now included the broken-down percentages

Figure 1: Add a scale bar and north arrow to the zoomed in map of the observatory site.
Has been added

144-5: Do the authors think that the drought conditions preceding the study could have impacted the results in any way?
Yes, in fact we mention this in the discussion, lines 406-413: "This may be a "memory effect" in the moisture regime of soil-vegetation systems from the severe drought of 2018 and low rainfall over the winter of 2018–19. Circumstantial evidence suggests that transpiration in trees and shrubs could have been suppressed during the study period. Kuhlemann et al. (2021) showed that normalised transpiration for the tree plot at the SUEO did not always respond proportionately to increases in atmospheric moisture demand (indexed through potential ET) implying the trees are conserving water and may be experiencing moisture stress at times (cf. Johnson et al., 2017). Additionally, the modelled transpiration response to the addition of run-on (Fig. 8) implies water limitation for trees inhibiting growth."

151: Missing words in this sentence - '… installed at the top of the flux towers…'
Revised

150-7: What is the fetch (or footprint) of each tower?
The tower on-site appr. 500-600 meters, the tower 6 km north of site appr. 700-800 m. Added in section 2.2.
What proportion of the study period is each tower measuring convective fluxes that are representative of the observatory site? This will depend on wind direction of course.
The tower on site is expected to be measuring close to representative conditions at all times, regardless of wind direction. This therefore is true for the calibration period, (as the tower on site went into operation the preceding summer). Also, we want to point out that we primarily use the eddy flux tower data to drive the model. The calibration and validation are mainly done by using other measurements on-site.

Section 2.2.: Throughout this section it would be helpful to know the make/model of all the sensors that were used.
Has been added.

161-4: How was soil sampled? How was water extracted? How was isotope analysis performed?
166: I recommend providing a brief overview of methods and then referencing Kuhlemann et al 2020b for more detail.
We have added more detail in the text, as well as in the supplementary material to make our paper more standalone. Also, full details are available with Open Access in Kuhlemann et al. (2021), *HESS*.

188-9: Is available moisture at the surface not used as well for partitioning available energy into convective fluxes?

That is correct, we have revised the sentence.

198: How is the vegetation rooting parameter obtained? Is it based on field measurements of rooting depth?
It is a calibration parameter, see Table 1, assuming an exponential root profile.

204: The calibration period is the period over which data were obtained from the site. Validation is qualitative. Is there 2020 data that can be used to validate?
Yes, there is SWC data but no soil isotope data. We have performed this validation step as per suggestion and performance is similar as during calibration in 2019. We have included this validation step in the revision.

206: The soil division in the model is unclear to me. Were soil surveys used to characterize the spatial distribution of soil types? If not, was there any verification of the assumed soil characteristics?
Many soil characteristics were used as calibration parameters and we made sure to stay within reasonable parameter ranges. We have clarified the part about the soil division in the manuscript.

Figure 3b: Is the local drainage network storm sewers? Is there any channelized flow through the site?
The local drainage network is topographically defined and shows the direction that water would flow in the model in case of overland flow. In our study, the local drainage network is really only relevant for the exploratory study described in sections 2.5, 3.6 and 4.4. For the green spaces that are the focus of the study, there is no overland flow.

239: I think you could use 'ground-to-atmosphere' instead of 'upwelling'.
We do think 'upwelling' is the correct term and continue to use it.

247-9: How common are "infiltration hotspots" at sharp interfaces between impermeable and permeable surfaces? In my experience, these areas are usually quite compacted, thereby preventing infiltration.
Exactly how common these infiltration hotspots are would be important to address in future upscaling studies. However, even if the immediate area next to an impermeable surface would be compacted, water would then continue flowing on top of this surface and infiltrate where compaction is less strong, still creating an infiltration hotspot, only further from the impermeable surface.

267-8: Were surface cracks visible to the field team?
No, they weren't. Another interpretation, based on subsequent data collected at shallower depths, is that water did not reach down to the sensors at 15 cm, while in the model the top layer starts at the surface and therefore registers any infiltration into the soil. This has now been added to the paragraph.

274-5: Are the authors referring to the measured or modelled SMC here? Layer 3 is not shown in Figure 2. It's not clear which data is being discussed.
It was mentioned in the following sentence that we were referring to the measured SMC. We have now rearranged the two sentences to make this clearer.

299-300: I think the authors should try to provide some interpretation of the poor fit in late June, late Aug and early Sep. Thanks for this suggestion.
Including the sap flow measurements was not meant to be a quantification of transpiration, what we wanted was a qualitative comparison of the variability. We have updated Figure 7 as per request of reviewer #2, adding an envelope showing the range of the measured sap flow for the individual monitored trees.

307-8: What do the authors mean by 'This compares with the 352 mm of precipitation during the calibration period.'? All the ET values are larger than P.
We have reformulated the sentence.

318-20: Is there some way to estimate E from these surfaces to see if they account for the missing amount?
Yes, with some assumptions about surface storage on buildings/sealed surfaces and evaporation rates, this would be possible but we would prefer to keep the focus of this paper on urban green space. We have revised the paragraph.

324-7: In layer 1, SWC is generally higher at the beginning and end of the study period with fluctuations (but lower baseline) in between. Why is the water age relatively high in April/May and Nov? Maybe the April soil water is old from the previous winter, but November receives a fair amount of precip. Some explanation of the distribution of ages at the beginning and end of the study period would be helpful.
In fall/winter much less water leaves the compartments, less transpiration, less evaporation, and when precipitation infiltrates into the soil there is therefore more water already there to be mixed with. You do see that when there is rainfall in fall, the ages go down, but not as extreme as in summer when much less water is available in the soil, making the impact of young water on the soil water ages bigger. We have made this clearer in the revision.

362-5: Would the authors recommend more, denser SWC measurements in urban soils?
That is one option if resources are available. Another idea would be to integrate the measurements over the soil profile and create virtual layers from that within the model domain.

391: Spelling error – trees.
Corrected

396: It's unclear what '… generally capturing processes adequately…' means. What were the criteria for evaluating this?
Visual comparison (plot 6) and KGE values.
On line 397, 'generally good reproduction' is used. Can the authors be more quantitative, e.g., measured and modelled isotope values were within x-x % of one another?
We have added NRMSE values in section 3.3 for layers 1 and 2, to complement the KGE values for layer 1. We have also reformulated section 4.3.
It's unclear what is meant by '… this simulation required…'.
Has been reformulated

399: Similar to the last comment, '… somewhat less successful…' seems too vague. Can the fit be assessed quantitatively? (As above)

399-400: I recommend being more explicit about the number of wetting fronts that the isotopic results reflected.
We have clarified this paragraph.

Figure 10: This is a nice conceptual figure.
Thank you.

Section 4.4: It might be worth mentioning that sometimes preferential flow pathways can form along impermeable-permeable surface boundaries. This would move water away from the area and potentially make it unavailable for green or blue water fluxes. Something to explore in future work perhaps. [I see the authors make this point later on line 454-6… excellent].

Section 4.5: Could the authors comment further on the role of grass in promoting green water fluxes? Many sustainability-focused landscape designers seem to be moving away from grass (or lawns), but perhaps the findings of this study are an argument in their favour.

We had already mentioned the study by Gómez-Navarro et al. (2021), which stated that a combination of turfgrass and trees can be beneficial in combating UHI. We have now also a reference to Litvak and Pataki (2016), showing how shading lowers the water demand for lawns, further indicating how a combination of trees and grass could be beneficial, with the trees providing shading for the grass.

**References**

Kuhlemann, L.-M., Tetzlaff, D., Smith, A., Kleinschmit, B., and Soulsby, C.: Using soil water isotopes to infer the influence of contrasting urban green space on ecohydrological partitioning, Hydrol. Earth Syst. Sci., 25, 927–943, https://doi.org/10.5194/hess-25-927-2021, 2021.

**Comment on hess-2020-640**
**Anonymous Referee #2**

Referee comment with author replies (post-revision) on "Quantifying the effects of urban green space on water partitioning and ages using an isotope-based ecohydrological model" by Mikael Gillefalk et al., Hydrol. Earth Syst. Sci. Discuss., https://doi.org/10.5194/hess-2020-640-RC2, 2021

Author replies in red

The manuscript "Quantifying the effects of urban green space on water partitioning and ages using an isotope-based ecohydrological model" written by Gillefalk *et al.* provides a set of insights for water partitioning in a complex urban landscape. They incorporated the use of water stable isotopes in precipitation and soil to verify the model capacity for partitioning water fluxes. Also, they use eddy flux and sap flow data to evaluate the model results. Despite their meticulous work, there are some concerns about data collection and applicability.

We thank Reviewer 2 for their suggestions and complementing our "meticulous" work. We address the data concerns that the reviewer raises below.

**Major Comments**

**Flux tower:**

Authors mentioned in lines 147-150 the use of another urban flux tower for a portion of the sampling period. It is important to highlight the fact that despite their similarities as "Urban Environments", the proportion of green spaces/buildings can affect considerably the model outputs. Also, the authors did not show a consistency analysis or a comparison for the period June – November in which both towers could be operating.

This is fundamental to considerer the fluxes as similar, equal, or different. Fluxes such as outgoing longwave and shortwave radiation, as well as water vapor can be affected and give different proportions. As an example, the 2-fold overestimation showed in Figure 7

with respect to sap flow data for April and May can be linked to the differences between flux towers without counting on the constant LAI effect.

The authors should ensure that this data set can be used by this other location.

* How different are the land covers within the tower's footprints?

* Are the fluxes for the period June-November equal/proportional/different?

* Do both towers have the same setup in terms of instrumentation?

We do clearly state the periods of each eddy flux tower at the beginning of section 2.2. From the summer 2018 until the end of the of the modelling period in Nov 2019 we use the data from the tower on site. Therefore, no differences shown in figure 7 can be explained by the use of data from the second tower. The tower located 6 km north of the study site is only used during the first part of the spin-up and any difference in land covers should therefore be negligible for the calibration period.

**Calibration:**

The manuscript is based on the application of a model which requires a calibration period. The authors mention the application of this procedure (Section 2.4). Despite the detailed description of the calibration procedure, two main questions remain unanswered:

* Which data (period and source) was used for the calibration?

* Did the authors apply a spin-up procedure (how long) or not?

This issue is important to assess possible trends or initial effects in the flux initial values.

We are slightly surprised by this comment as both these questions were clearly answered in the originally uploaded manuscript, specifically lines 158-160 and lines 203-204 (lines 162-164 and lines 205-206 in the latest version).

**LAI and sap flow:**

During the modeling procedure, the authors used a constant Leaf Area Index per cover. This can be true for the grassland depending on the species but the effects in trees and shrubs fluxes can be important. The application of this assumption triggered important consequences for the model results which end up with the overestimation of transpiration fluxes during the first part of the year (Figure 7 – April and May).

However, the lack of sap flow data in shrubs affects the reliability of the fluxes from this cover.

As noted in the response to Reviewer #1, including the sap flow measurements was not meant to be a quantification of transpiration, what we wanted was a qualitative comparison of the variability. The overestimation only is visible in April and was exaggerated by our use of an average sap flux measurement from all sensors. As suggested below by the Reviewer, we have now included a new version of the figure showing the range of measured values.

**Urban karst**

Along with the discussion, the authors mention the term "urban karst" given by Bonneau et al. (2017) which affects the water fluxes and redistribution by the preferential flow. Taking into account the heterogeneity of the subsurface on the sampling area (Lines 134-135) where the "subsurface is heavily impacted by human activities, and in places

has an added layer of up to 50–180 cm of debris", how does the preferential path flow form by these debris affects or potentially affects the soil water age estimations?

Our comments were perhaps misleading here. The study site includes significant areas of made ground with some subsurface building rubble, though the upper 1-2 m of soil is generally now quite well developed and no debris relevant for our study was found during field work. We have now added a sentence on this in the manuscript.

**Tree with run-on**

How does the "tree with run on water" compare against the transpiration of the tree(s) sap flow measured in those pixels? This will support the affirmation given by the authors about the model performance and side effects of nearby impermeable land covers (e.g, buildings, pathways).
Unfortunately, sap flow was not measured in the pixels where we explored effects of tree with run-on, so we are unable to do this.

**Minor Comments**

Does the soil water content measurements were calibrated with soil samples along the sampling period?
Yes, the TDR probes were calibrated to local data at installation.

The authors mention the use of a German Weather Station that "records essentially the same rainfall" (Line 156). Can the authors provide the values?

We have now clarified this in the manuscript.

What are the urban tree species sampled for this manuscript (Lines 159-161)? Can the authors provide more information about the individual trees sampled (e.g, diameter, species, height, etc)?
The tree species sampled were maple, elm, plane and oak. We have added this in the manuscript. Full details can be found in the now published open access article Kuhlemann et al. (2021), *HESS*.

The paragraph between lines 167 to 175 describes the results obtained from the data collection described in the previous methodological sections. Consequently, this should be in Results and not in Methods and Material.
Again, the collected data are part of a related data-driven study (Kuhlemann et al. 2021, *HESS*) and hence need to be presented as background context for this modelling study, rather than new results.

The authors mention the use of Nash-Sutcliffe efficiency (NSE) as objective functions (Line 216). However, across the manuscript, there is only one reference to NSE in a broader context (Line 234) with no reference to the results of this analysis and neither in the supplemental material. The authors only mention Kling-Gupta Efficiency in detail (e.g, Line 272, 282). What happened with the NSE analysis?
We chose to focus on the KGE values as we deem them better suited for evaluating model performance of soil water content, as there is less primacy on the simulation of peak values alone. But we have now added the NSE values in the text in section 3.2.

- The authors should follow the recommendations given by Knoben *et al.* (2019) when using Kling-Gupta Efficiency analysis in models.
During the modelling the aim was to maximise KGE, no benchmark was explicitly set.

- The authors should add more information about the results using the NSE. Also, adding the respective equations as for KGE.

See above for comment on NSE. We present the equation for KGE as it is a lesser-known performance metric and since we put most focus on it.

- It is necessary to add more information about the sampling processing (precipitation and soil samples). The following questions must be answered:

\* How were soil samples collected?

\* When collected, how long after a rain event the soil samples were taken?

\* Which soil water extraction procedure was applied?

\* What method/equipment/laboratory performed the stable water isotope analysis?

We have added more details in the manuscript and in the supplementary material to make our paper more standalone. However, full details are available with in Kuhlemann et al. (2021), *HESS*.

**Recommendations**

The authors can use boxplots in "Figure 10: Soil layers" instead of bars. In this way, the reader can have a better idea of the data distribution for each layer/cover.
We have considered this suggestion for the revision, but using boxplots for the water balance does not work well visually. The spread of values is so low for most components and vegetation types that boxes turn into lines. Also, we note Reviewer #1 liked the summary plot as it stands.

The authors can add the transpiration envelop in Figure 7. This will allow the readers to have a better notion of the temporal flux variability.
This was a very good suggestion. We have added an envelope for the measured sap flow, ranging from min to max values for each day and replaced the old version of this figure in the manuscript.

**References**

Bonneau, J., Fletcher, T. D., Costelloe, J. F. and Burns, M. J.: Stormwater infiltration and the 'urban karst' – A review, J.Hydrol., 552, 141–150, 10.1016/j.jhydrol.2017.06.043, 2017.

Knoben, W. J. M., Freer, J. E., and Woods, R. A.: Technical note: Inherent benchmark or not? Comparing Nash–Sutcliffe and Kling–Gupta efficiency scores, Hydrol. Earth Syst. Sci., 23, 4323–4331, https://doi.org/10.5194/hess-23-4323-2019, 2019.

Kuhlemann, L.-M., Tetzlaff, D., Smith, A., Kleinschmit, B., and Soulsby, C.: Using soil water isotopes to infer the influence of contrasting urban green space on ecohydrological partitioning, Hydrol. Earth Syst. Sci., 25, 927–943, https://doi.org/10.5194/hess-25-927-2021, 2021.

**Summary of relevant changes, excluding language improvement and minor clarifications:**

**Data collection**

We've added more detail on soil sampling, water extraction, isotope analysis etc. in section 2.2 and in the supplement.

**Energy balance**

We've addressed comments from reviewers and editor regarding flux tower and energy balance. Lines: 150-152, 191, 333-336

**Figures:**

Fig. 1. Added North arrow and scale bar.

Fig. 7. Added envelope showing the measured sap flow, ranging from min to max values for each day.

Fig. 10. Made the symbols consistent ($T$, $E_I$, $E_S$).

**NSE values**

We've added the NSE values for the final run to the results section. Lines: 285-287

**Soil water isotopes**

We've calculated and added the performance of soil water isotope modelling using NRMSE add revised the discussion regarding them. Lines: 243, 305-307, 422-427

**Tree species**

We've added the tree species where sap flow measurements were made. Line: 166

**Validation**

We've performed and included validation using soil water content data from 2020. Lines: 251-253, 291-292, 385-386

**Vegetation choice for promoting green water fluxes**

We've added additional reference in the discussion. Lines: 494-496

**Water age**

We've elaborated on the simulation of the soil water ages. Lines: 345-348.